Resource

# A comprehensive resource for retrieving, visualizing, and integrating functional genomics data

Matthias Blum[1],*, Pierre-Etienne Cholley[1],*, Valeriya Malysheva[1], Samuel Nicaise[1] , Julien Moehlin[1], Hinrich Gronemeyer[1,2,3,4], Marco Antonio Mendoza-Parra[1,2]

**The enormous amount of freely accessible functional genomics data is an invaluable resource for interrogating the biological function of multiple DNA-interacting players and chromatin modifications by large-scale comparative analyses. However, in practice, interrogating large collections of public data requires major efforts for (i) reprocessing available raw reads, (ii) incorporating quality assessments to exclude artefactual and low-quality data, and (iii) processing data by using high-performance computation. Here, we present qcGenomics, a user-friendly online resource for ultrafast retrieval, visualization, and comparative analysis of tens of thousands of genomics datasets to gain new functional insight from global or focused multidimensional data integration.**

## Introduction

The functional genomics data available to date in public repositories are still largely underexploited and insufficiently incorporated in ongoing biomedical research. Indeed, the combination of a variety of molecular biology approaches with massive parallel sequencing gives rise to exponentially increasing amounts of functional genomics data concerning protein–DNA interactions and global epigenetic landscaping (e.g., chromatin immuno-precipitation evaluated by massive parallel DNA sequencing [ChIP-seq]), chromatin accessibility (e.g., assay for transposase-accessible chromatin with high-throughput sequencing [ATAC-seq]), and gene expression (e.g., RNA-seq), as well as the three-dimensional chromatin organization (e.g., Hi-C) and new technologies for single-cell functional genomics. Estimates predict

that in 2025 between 2 and 40 Exabyte of genomics information will be available for analysis (Stephens et al, 2015), hosted in a variety of public repositories, such as the Gene Expression Omnibus (GEO) and others (Grabowski & Rappsilber, 2019). The value of this wealth of functional genomics data is enormous, as it concerns fields ranging from development and cell biology to (patho)physiology, precision medicine, discovery of biomarkers, and therapeutic targets and has the promise to get towards an understanding of the molecularly encoded communication networks that are at the basis of living cells, organs, and individuals.

However, one of the caveats in interrogating and integrating publicly available data is that it requires computational biology expertise as well as major computing resources, which are available at major centers but scarce in medium/small size laboratories. Indeed, for optimal reuse, it is essential to reprocess public data under standardized conditions and to evaluate their quality to exclude low-quality or potential artefactual data, which could generate bias and lead to improper or wrong data interpretation. To address data quality, we have previously developed a quality control system for functional genomics data (Mendoza-Parra et al, 2013b), which has been used for qualifying at present more than 82,000 publicly available enrichment-related datasets; this quality assessment database comprises ~70% of all publicly available ChIP-seq assays generated worldwide.

Starting from this quality assessment, we have developed a user-friendly suite of big data analysis tools—qcGenomics (http://ngs-qc.org/qcgenomics/)—a publicly available resource to retrieve datasets of user-defined quality according to a multitude of query options and visualize them through a dedicated genome browser. More importantly, we have implemented solutions for both global and local comparative analyses to study from two up to several hundreds of datasets to reveal, among others, common features/signatures. Thus,

[1]Department of Functional Genomics and Cancer, Institut de Génétique et de Biologie Moléculaire et Cellulaire, Equipe Labelisée Ligue Contre le Cancer, Illkirch, France [2]Centre National de la Recherche Scientifique, UMR7104, Illkirch, France [3]Institut National de la Santé et de la Recherche Médicale, U1258, Illkirch, France [4]Université de Strasbourg, Illkirch, France

Correspondence: mmendoza@genoscope.cns.fr; hg@igbmc.fr
Matthias Blum's present address is European Molecular Biology Laboratory, European Bioinformatics Institute, Wellcome Genome Campus, Cambridge, UK
Pierre-Etienne Cholley's present address is Computational Systems Biology Infrastructure, Chalmers University of Technology, Gothenburg, Sweden
Valeriya Malysheva's present address is Functional Gene Control Group, Epigenetics Section, Medical Research Council London Institute of Medical Sciences, London, UK and Institute of Clinical Sciences, Faculty of Medicine, Imperial College, London, UK
Samuel Nicaise's present address is University Hospital, Unité de Génétique Moléculaire, Strasbourg, France
Julien Moehlin's present address is Unité Mix de Recherche (UMR) 8030 Génomique Métabolique, Genoscope, Institut François Jacob, Commissariat à l'Énergie Atomique (CEA), Centre National de la Recherche Scientifique (CNRS), University of Evry-val-d'Essonne, University Paris-Saclay, Évry, France.
* Matthias Blum and Pierre-Etienne Cholley contributed equally to this work
Marco Antonio Mendoza-Parra's present address is UMR 8030 Génomique Métabolique, Genoscope, Institut François Jacob, CEA, CNRS, University of Evry-val-d'Essonne, University Paris-Saclay, Évry, France

without the need of collecting and reprocessing the data, non-specialist users will be able to interrogate large amounts of functional genomics data, visualize enrichment patterns or identify, for instance, co-occurring binding patterns from a multi-profile comparison. Importantly, users can upload their own data—without having to install additional software—to visually compare it with those available in the public domain.

## Results

### qcGenomics: a web-access solution for an intuitive interaction with functional genomics data released on the public domain

We previously established an automated pipeline to download and realign raw datasets from the sequence read archive (SRA) to provide global and local quality assessments of large amounts of functional genomics data (Fig 1). This generated a public database (http://ngs-qc.org/database.php) in which quality indicators provided by the "next generation sequencing quality control (NGS-QC) Generator" (Mendoza-Parra et al, 2013b) are currently associated with >82,000 ChIP-seq and similar enrichment-related datasets, as well as to long-range chromatin interaction data (Hi-C and related; http://ngs-qc.org/logiqa) (Mendoza-Parra et al, 2016). We have now implemented a dedicated data portal (termed "NAVi" for Nucleic Acid Viewer) that allows to query publicly available data by combining intuitive keywords such as cell/tissue type, model organism, target molecule, accession numbers, associated quality score, author names, and keywords in the title or abstract of a corresponding article. As a consequence, NAVi displays the user-selected query in a table format where further information, including the source of the public data (GSM ID) and the number of total mapped

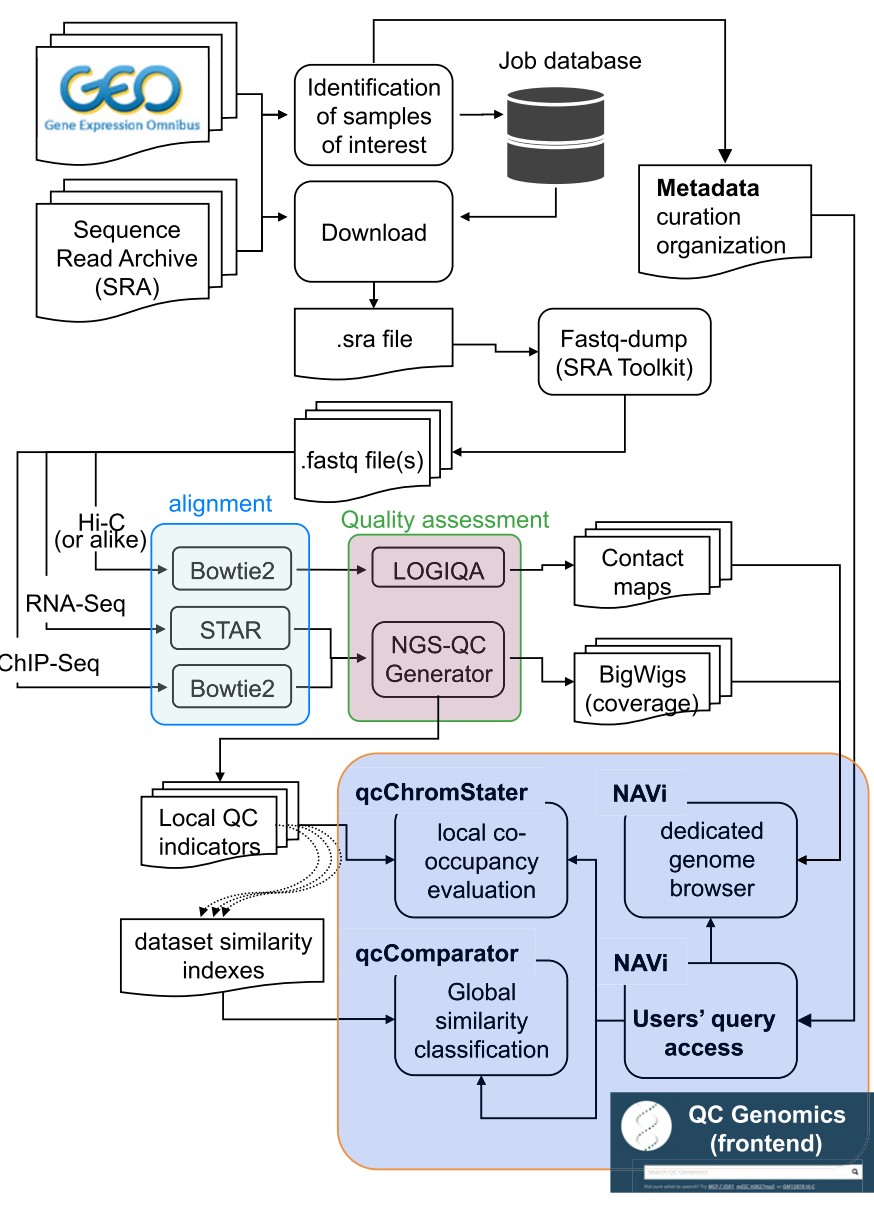

**Figure 1. Schematic illustration of the qcGenomics workflow.**
An automated web routine identifies and indexes novel datasets available in GEO and SRA and stores such indexes in a database. A job scheduler dispatches tasks to computational nodes. A job consists of downloading SRA files from the SRA database, converting SRA files to FASTQ files, aligning raw reads, and controlling the quality of mapped reads. Before each NGS-QC database update, similarity indexes used by qcComparator are generated for all pairwise comparisons of experiments, and experiments are enriched with annotated metadata.

reads. In addition, users can select datasets of interest and visualize their enrichment patterns with the dedicated NAVi genome browser. Notably, NAVi provides flexibility by displaying on-demand HiC contact maps and ChIP-seq enrichment coverage in a single view (Fig 2), thus providing optimal conditions for comparative studies and intuitive searches. In the illustrated example, Hi-C long-range interaction maps in the surrounding of the SOX2 locus are displayed together with the enrichment patterns for the histone modifications H3K4me3, H3K4me1, and H3K27ac. Note that two (or more) analyses can be displayed; in the example, HiC and ChIP-seq patterns for a given region in embryonic stem cell (ESC) datasets are presented together with those of neuroectodermal cells.

To facilitate retrieval of public data, queries in NAVi can contain multiple terms surrounded by double quotes to search for an exact group of words and can be combined with a plus sign to combine the results of multiple independent queries (e.g., "MCF-7 H3K4me3 ChIP-seq + MCF-7 ATAC-seq"). As an outcome, NAVi provides a list of all datasets matching the query, accompanied by several descriptors, including the associated model organism, type of biological source (cell/tissue type), target molecule, number of mapped reads, quality score, and a direct link to the original accession ID (Figs 3A and S1).

Data retrieved by NAVi can be subsequently analyzed and visualized by a number of additional tools. First, a fraction (or all) of the retrieved data can be selected and displayed with the NAVi genome browser (Fig 3B). This type of analysis is gene/genomic region–centered and designed for visual inspection.

In addition to classical browser option (zoom-, gene-, and genome coordinate–based searches, genome walking/jumping), NAVi displays enrichment quality patterns (described as local quality control regions or "local QCs") in a heat map bar plot format. In addition, it can display local read coverage and/or precomputed MACS2 (Zhang et al, 2008) binding-site predictions.

### Exploring public datasets for ultrafast comparative analysis and discovery of novel features

NAVi can be used for a variety of issues, ranging from data variability to comparative data analysis which can provide novel testable hypotheses. This is illustrated for the ERG transcription factor (TF) binding to the WNT7B gene in the normal prostate epithelial cell line RWPE-1 (Fig S2). The four datasets show vastly different signal-to-noise ratios, with two (GSM2195110 and GSM2195106) revealing no apparent binding in this region, whereas small peaks can be seen in GSM2195103 with the same antibody. In contrast, GSM927071 reveals strong peaks in the first and third exon of WNT7B with another antibody. Interestingly, the top three ChIP-seq profiles were constructed from 23 to 29 million mapped reads, whereas GSM927071 originates from about 15 million mapped reads. Using the latter as a guide, small peaks can be discerned also in the other four experiments for the peak within the first WNT7B in intron. Thus, within a few seconds the user can select and compare different datasets to search, find evidence, or gain confidence for a speculated binding event of a TF and validate this finding with gene-specific approaches.

Comparing RWPE-1 data with those of the TMPRSS2-ERG fusion-positive VCaP prostate cancer cells provides even information about those genes which have become under the control of the misexpressed ERG in the cancer cells. Indeed, although no significant ERG binding is apparent at the WNT2 upstream region in RWP1 cells,

ERG binding close to the transcriptional start site is readily seen in VCaP cells (Fig S3). This suggests an acquisition of ERG binding and (de)regulation of WNT2 expression in the cancer cells which acquire the TMPRSS2-ERG fusion.

Finally, combinatorial events, such as TF binding and histone modifications (Fig S4), can be rapidly monitored. The example displays the previously reported (Ross-Innes et al, 2010) co-binding of estrogen and the retinoic acid receptors in MCF7 cells. Obviously, these data can be combined with other data types, such as global run-on with high-throughput sequencing (GRO-seq), formaldehyde-assisted isolation of regulatory elements analyzed by high-throughput sequencing (FAIRE)/ATAC-seq, methylated DNA immunoprecipitation sequencing (MeDIP-seq), and many more, in a user-defined manner. Thus, NAVi is a user-friendly sophisticated tool for data search, retrieval, and rapid comparative analysis on a visual basis by integrating very different functional genomics datasets.

Although having a genome browser linked to virtually all public ChIP-seq or related datasets is extremely useful, performing visual gene/genomic region–centric analyses in hundreds of datasets at the same time is impossible. For this reason, we developed computational solutions to compare multiple datasets in an unbiased, not gene-centric manner. The two resulting tools, qcComparator and qcChromStater, provide global and local multi-profile comparisons for large numbers of profiles.

### qcComparator—similarity landscaping of hundreds of public datasets

To find similar landscapes in large numbers of ChIP-seq and related datasets, qcComparator compares genome-wide enrichment patterns and generates a quantitative global similarity readout. In contrast to previous efforts, which relied on peak caller predictions (Devailly et al, 2016), we based the comparison on the local quality assessment along the genome (local QCs) computed by our quality control strategy (Mendoza-Parra et al, 2013b), which is insensitive to peak shapes.

Briefly, local QCs quantify the resilience of signals in all 500-bp bins of the genome, when only 50% of the mapped reads are used to reconstruct the enrichment profile. For TF ChIP-seq profiles, we have previously demonstrated that local QC regions correspond to highly confident binding events, which are generally identified by peak calling (Zhang et al, 2008; Mendoza-Parra et al, 2013a; Thomas et al, 2017). As local QC annotations are computed with the same computational strategy (Mendoza-Parra et al, 2013b), they are insensitive to the type of enrichment pattern, whereas peak calling algorithms perform differently for "sharp" or "broad" enrichment patterns. Datasets selected for display with the NAVi browser are systematically accompanied by local QC heat maps and compiled read counts (Fig 3A and B) or, optionally, MACS peaks. We used these local QC enrichment landscapes as basis for multi-profile comparisons and computed pairwise similarity indexes for all datasets in each model organism (>37,400 for human and >35,100 for mouse profiles). This precomputing step was essential to facilitate rapid access to several hundred profiles for multiple comparisons. As a consequence, users can query for the data of interest with the NAVi portal (Figs 3A and S1) and perform immediately global comparative analyses with qcComparator. The tool displays similarity indices for all queried datasets in a matrix format and performs data classification by clustering.

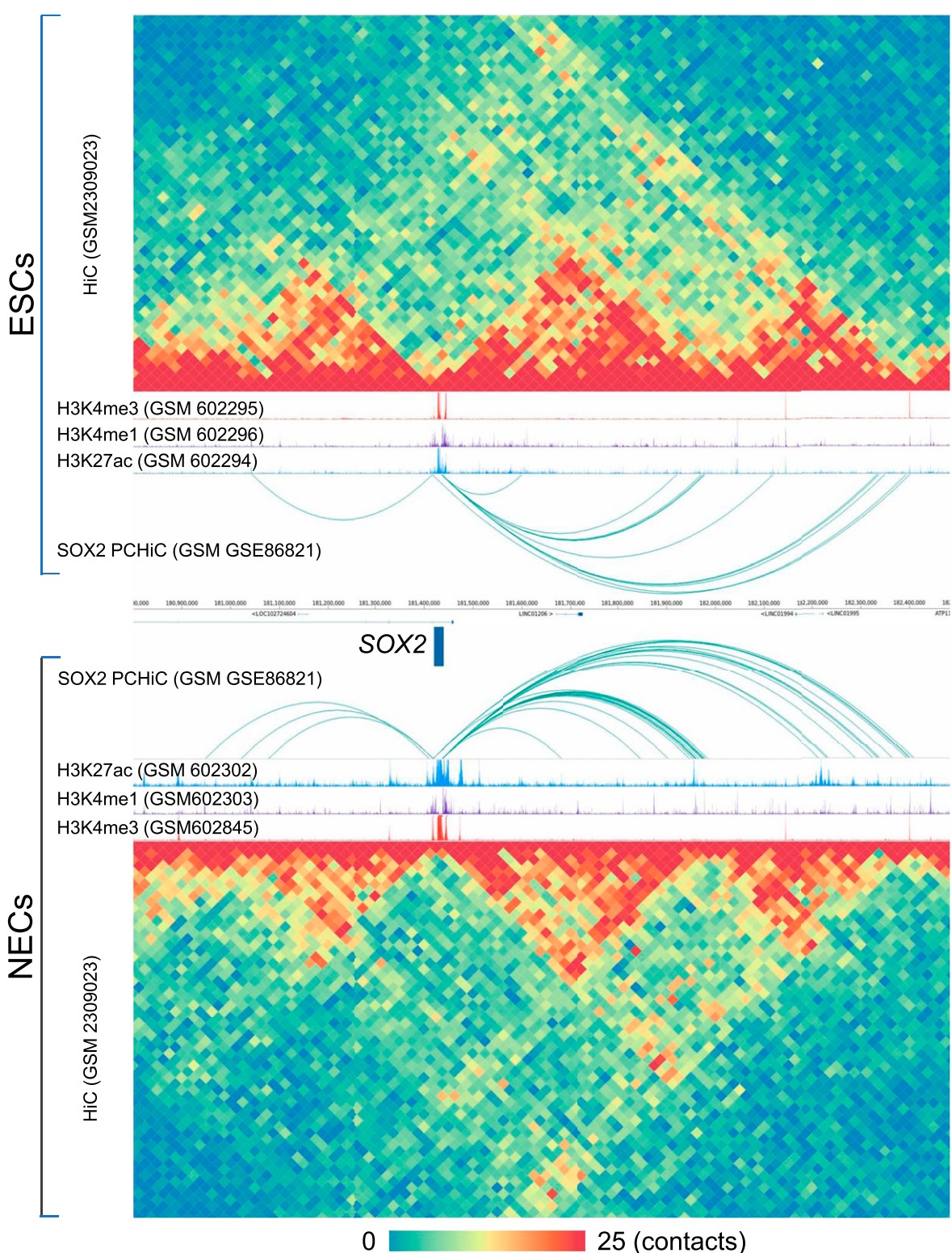

**Figure 2. Example of a comparative visualization with qcGenome browser NAVi of histone modification tracks and HiC-derived illustrations for the Sox2 region.**
Shown are the HiC-derived 3D chromatin organization illustrating the topologically associated domains; tracks for the enhancer/promoter-associated histone modifications H3K4me3, H3K4me1, and H3K27ac (track height 50 reads); and chromatin interaction loops from Promoter Capture Hi-C (PCHiC) experiments, illustrated as arcs. The displays are in mirror image order for human ESCs above (top panels) and ESC-derived neuroectodermal cells below (bottom panels) the genome coordinates and the genes in this region. The color code for the chromatin contact map given at the bottom.

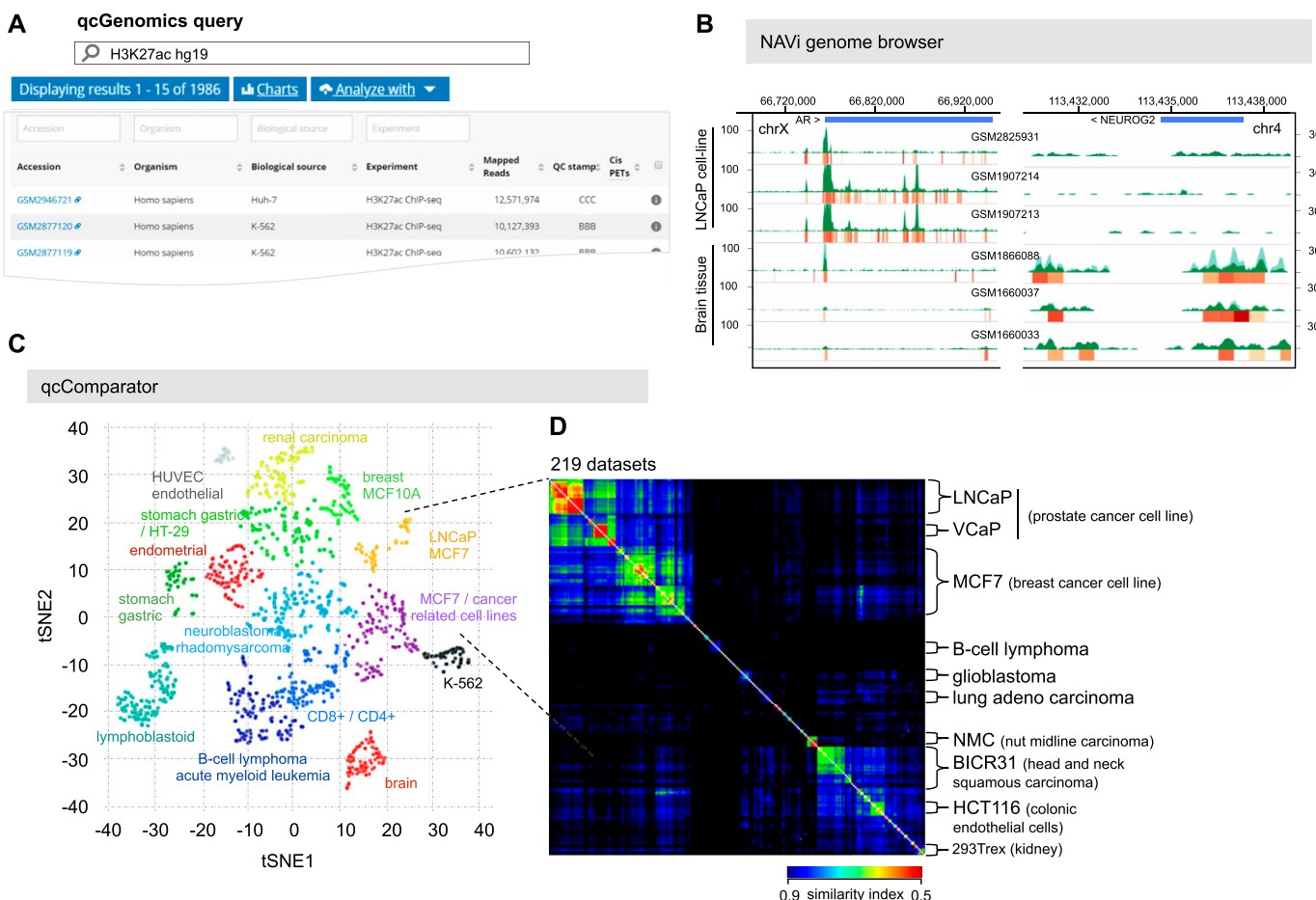

**Figure 3. Example of a qcGenomics query illustrating the retrieval of datasets by NAVi, their visualization with the embedded genome browser, and the datasets comparisons on the basis of their global enrichment patterns by the tool qcComparator.**
**(A)** 1,986 human public datasets corresponding to ChIP-seq assays targeting the histone modification H3K27ac were retrieved with NAVi by the query "H3K27ac hg19". This information is displayed in a table, where further elements, including the biological source and the total mapped reads, and also their associated global quality score are provided. **(B)** A subset of datasets retrieved in this query are visualized with the embedded NAVi genome browser. Here, local quality-controlled regions (local QC) are displayed as a heat map bar plot (orange gradient bars), accompanied by their read count enrichment pileups (green). Note the differential enrichment pattern for H3K27ac around the androgen receptor gene in LNCaP cells relative to that observed in brain tissue and the inverse when visualizing Neurogenin 2 (NEUROG2). **(C)** All retrieved datasets in (A) were compared by computing pairwise similarity indices (Tanimoto) and classified with the t-SNE strategy. This analysis provides a sample stratification, which was further correlated with their related cell/tissue origin. **(D)** A detailed view of two of the t-SNE–identified clusters, namely, LNCaP and MCF7/cancer–related cell lines (219 datasets) performed by qcComparator is displayed. Here, dataset similarity indexes (Tanimoto) are displayed in a heat map square matrix. Note that LNCaP and VCaP cell–related datasets–both originated from prostate cancers–present a higher degree of similarity when compared with other types of cancer cell lines.

Global comparisons, as that performed by qcComparator, allows for instance to classify datasets issued from multiple cell/tissue types. To reveal its performance, qcComparator was challenged to stratify human ATAC-seq landscapes. ATAC-seq chromatin accessibility landscapes are well established to reveal cell identity and search the cell-of-origin in the case of cancer (Polak et al, 2015; Corces et al, 2016). Indeed, qcComparator clustered the ATAC-seq datasets of CD34+, CD4+, and CD8+ hematopoietic cells into four main clusters; interestingly, peripheral blood CD4+ cells, most of which came from a study of inter-individual variations (Qu et al, 2015) segregated into two gender-independent clusters of apparently dissimilar chromatin accessibility landscapes (Fig S5). To illustrate the potential of qcGenomics to compare hundreds of profiles, we queried in NAVi the human datasets for the histone H3K27 acetylation (Fig 3A), which is generally considered to mark

active enhancer regions (Nord et al, 2013). We retrieved 1,986 datasets, which were stratified on the basis of their global similarity index (t-distributed stochastic neighbor embedding or t-SNE [(van der Maaten & Hinton, 2008)]). The similarity-based clustering coincided in most cases with the associated cell/tissue type (Fig 3C). To further enhance the detection of dis/similarities between datasets, we focused our attention on the 219 dataset clusters containing the LNCaP prostate and MCF7 breast cancer cell lines (Fig 3D). The similarity index heat map (implemented in qcComparator) revealed a higher similarity between two prostate cancer cell lines (VCaP and LNCaP) than to the MCF7 breast cancer and other more divergent cancer cell lines in this super-cluster. The matrix display of qcComparator corresponds to an interactive 2D map, such that datasets can be highlighted on the basis of their target molecule, quality, or cell/tissue type. In addition, heat map

contrast can be altered by the user who can also zoom into areas to sub-select regions in the matrix. Finally, it is also possible to switch between Tanimoto and Dice similarity distance metrics, among other options (Fig S6).

To further increase versatility, the qcGenomics tool was designed to be used in an iterative manner. To illustrate this point, the LNCaP and brain tissue–related datasets, which appeared in two distinct clusters in the t-SNE analysis (Fig 3C), were selected and compared on the basis of the H3K27ac enrichment patterns at the androgen receptor gene and the neuronal TF NEUROG2. As expected, NAVi depicted clearly different enrichment patterns for these cell type–selective TFs, also supporting the global classification performed on the basis of the similarity index of the computed local QCs (Fig 3B). Thus, qcComparator enables the user to compare large amounts of datasets and define their global similarity, as a basic for refined scrutiny at particular loci/genes.

### qcChromStater—chromatin state analyses with hundreds of datasets

The function of a particular region of the genome is influenced, if not determined, by the multiplicity of events (e.g., TF binding) and modifications (e.g., epigenetic modifications, accessibility, and structure of chromatin) occurring at and around this region; hence, defining the chromatin state at the time "t" in a given cell/tissue system. Therefore, comparative analyses of local regions over large numbers of datasets can provide functional insight and identify critical players or the interplay between specific TFs, chromatin modifications, and signaling pathways. In this respect, several tools for chromatin state analyses have been developed (ChromHMM, [Ernst & Kellis, 2012], diHMM [Marco et al, 2017], and chromswitch [Jessa & Kleinman, 2018]). However, performing such analysis with several hundred datasets remains challenging, as all data need to be pre-processed for a given tool, each of which has limitations for processing large numbers of data.

To overcome these limitations, we developed qcChromStater, which infers combinatorial states by comparing local QC genomic regions associated with the datasets retrieved by NAVi. Although the analysis is performed genome wide, the outcome is displayed in a gene-centric manner, such that users can query for co-occurring events at the transcription start site (TSS), the gene body, the transcription end site (TES), and outside the coding region (proximal and distal regulatory element) (Fig 4B; users can modify distal and proximal response element boundaries). Fig 4A illustrates a query in NAVi for a combination of epigenetic modifications (H3K4me3, H3K27me3, and H3K27ac), binding of the histone acetyltransferase and transcriptional co-activator P300 and transcription (RNA-seq) for MCF7 breast cancer cells. This query retrieved 170 datasets which were sent to qcChromStater for chromatin state prediction. This analysis identified 18 states (minimum coverage per genomic region is set to 5% of the total accumulated local QCs) with associated putative functionality including active enhancer regions state ("s01" H3K27ac enrichment) preferentially found at the proximal response elements and at the gene body which can harbor enhancers (Birnbaum et al, 2012; Lee et al, 2015), actively transcribing genes (state "s02": RNA-seq enrichment) detected at gene body and TES, transcribing genes with active promoter markers (state "s04": RNA-seq+ H3K27ac+H3K4me3 and state "s05": RNA-seq+H3K27ac), but

also repressed coding regions (state "s03": H3K27me3). Although qcChromStater infers combinatorial states, their association to functional annotations remains to be performed by the user. For this, the tool provides a field ("Functional Annotation") to add this information, which is then used by qcChromStater for further state stratification and association to coding regions (Fig 4C). Indeed, as illustrated in Fig S7, qcChromStater can extract the list of annotated genes associated to each of the user-defined states and filter them on the basis of their association to gene regions (TSS, Gene Body, and TES but also proximal or distal regulatory element association).

As part of qcGenomics, it is possible to verify such predictions by using the NAVi browser for their visualization. For example, the combinatorial state "s03" (considered a "repressive" state) presents ~30% of H3K27me3-associated local QCs at gene body regions ("Gene Coverage" heat map in Fig 4C). By clicking a given field in the heat map (Gene body), the list of genes associated to this combinatorial state is displayed. This list includes broad gene clusters, such as the carcinoembryonic antigen gene family composed by the pregnancy-specific glycoprotein (PSG) genes (Figs 4D and S8). In a similar manner, the combinatorial state "s04" (active promoter) comprises multiple active genes, exemplified here by the growth-regulating estrogen receptor binding 1 gene (Fig 4E).

Taken together, qcChromStater is unique in that chromatin states in a broad sense—including features such as TF binding or chromatin accessibility—can be inferred instantaneously from any public dataset in the qcGenomics database without any prior downloading or reformatting.

## Discussion

Several databases have been described for visualization and download of certain types of functional genomics data (Griffon et al, 2015; Sanchez-Castillo et al, 2015; Albrecht et al, 2016; Devailly et al, 2016; Mei et al, 2017; Cheneby et al, 2018; Dreos et al, 2018). However, these databases are limited to a few thousand datasets or dedicated to certain tissues. Also, several tools have been described to visualize genome interactions (Zhou et al, 2013; Kerpedjiev et al, 2018; Robinson et al, 2018). However, none of those provides the integrative capacity, variety, and versatility of qcGenomics with its huge precomputed database or with quality assessment indicators.

Importantly, qcGenomics harbors multiple tools for genome-wide analyses of a large variety of different genomics-based technologies. Wherever possible, we have integrated the various tools, such that data obtained in one can be transferred to another tool for further analysis. This is the case for the central data retrieval unit (NAVi) which dispatches tasks to qcComparator or qcChromStater, and the results can be fed back to NAVi for further (e.g., gene centric) analysis. In addition, qcGenomics harbors previously described tools, including the NGS-QC Generator (Mendoza-Parra et al, 2013b) for users to quality assess their own data, and TETRAMER (Cholley et al, 2018) to model gene regulatory networks and identify master gene in patho/physiological cell fate transitions as autonomous elements; all these tools are freely accessible online.

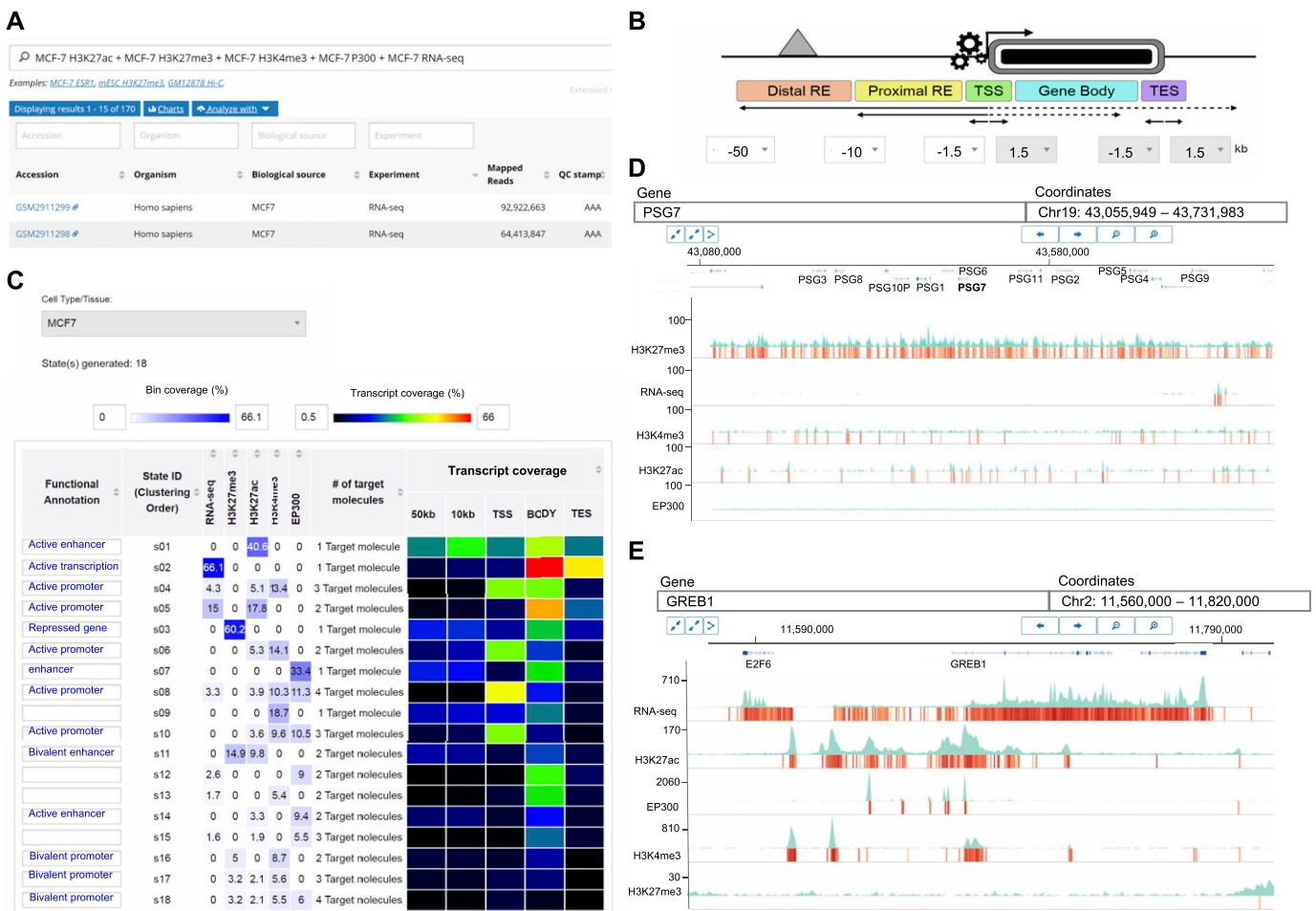

**Figure 4. Chromatin state analysis over large number of public datasets with qcChromStater.**
**(A)** Query concerning several histone modification marks, the co-activator facto P300, and RNA-seq datasets for the MCF7 cell line has been performed with NAVi. This query identified 170 datasets which has been compared in a local context (500-nt bins) by performing a combinatorial analysis. **(B)** Schematic representation of the gene-centric analysis performed by qcChromStater over the combinatorial states retrieved genome-wide. **(C)** The combinatorial analysis identified 18 different states (State ID: s01-s18), which were further stratified on the basis of their gene coverage (TSS, Gene Body, TES, and proximal/distal regulatory element regions defined at 10 and 50 kb, respectively). In addition, the functional annotation fields were filled on the basis of the knowledge corresponding to the different combinatorial states (e.g., "active enhancer" for regions enriched for the histone modification mark H3K27ac; or "repressed gene" for those associated to the H3K27me3 modification). qcChromStater allows to retrieve the genes related to each of the chromatin states, which can be verified by displaying the enrichment patterns with the NAVi genome browser. **(D)** Visualization of the PSG gene cluster associated to the state "s03" (repressed gene) in (C). Note that the histone modification mark H3K27me3 is enriched over the entire cluster, confirming the chromatin status predicted by qcChromStater. **(E)** Visualization of the growth-regulating estrogen receptor binding 1 genomic region, associated to the stat "s04" (active promoter) in (C). Note the strong enrichment patterns for the RNA-seq, H3K27ac, EP300, and H3K4me3 datasets, whereas there are very weak signals for the repressive mark H3K27me3.

By centralizing and uniform processing of datasets from various repositories, qcGenomics provides the most up-to-date and comprehensive online resource for public sequencing data generated by numerous methods from multiple organisms and biological sources. Without specialist computational skills, researchers can use qcGenomics to (i) find experiments of interest; (ii) visualize published genomic data; (iii) compare hundreds of high-quality profiles; (iv) infer functional chromatin states by integrative analysis of hundreds of datasets, and (v) analyze their own data in the context of the entire qcGenomics database. Currently, qcGenomics comprises 52,671 ChIP-seq, 5,457 chromatin accessibility (ATAC-seq, DNase-seq, formaldehyde-assisted isolation of regulatory elements analyzed by high-throughput sequencing [FAIRE-seq]), and 763 Hi-C experiments, and is regularly updated to integrate recently published experiments and cover new sequencing methods. The ambition of qcGenomics is to

enable scientists in the biological/biomedical/translational research without specialist informatics skills to extract information and generate hypotheses (see Figs S3 and S5), or test a hypothesis and compare their own data with the huge amount of data that have been accumulated in the past years. It should be pointed out that datasets of very divergent quality exist in the public domain; consequently, qcGenomics provides the user with the option to filter datasets for their robustness before performing the analysis. Moreover, the modular structure of the qcGenomics tools facilitates to use them in a sequential order and to switch from a large number of data to a few selected datasets of interest, or to switch from global and gene-centric analyses and back. Finally, the visualization options include chromatin interaction loops, which within the limits of resolution of the corresponding technologies, can inspire hypotheses about the cooperation of TFs, chromatin accessibility, and

modification patterns relative to the functionality of the genes in this genomic region.

# Materials and Methods

### qcGenomics access and documentation

Direct online access to the entire suite of tools is provided via the qcGenomics platform (http://www.ngs-qc.org/qcgenomics/). NAVi is the point of entry for searches and display of the results, as well as subsequent analysis of selected datasets by qcComparator and qcChromStater. NAVi provides an online tutorial ("Start the tour") and specific manuals are available online for each tool under "Documentation" of the corresponding page (http://www.ngs-qc.org/navi/index.php; http://ngs-qc.org/comparator/index.php; http://ngs-qc.org/chromstater/index.php). qcGenomics interrogates the content of our previously described NGS-QC Generator (http://ngs-qc.org/database.php) (Mendoza-Parra et al, 2013b) and LOGIQA (http://ngs-qc.org/logiqa/) (Mendoza-Parra et al, 2016) databases. The qcGenomics platform provides also direct access to TETRAMER (http://www.ngs-qc.org/tetramer/), a Cytoscape app which allows the reconstruction of transcriptional gene regulatory networks from user-provided expression data for temporally evolving or stepwise differential (patho)physiological states.

### Data collection

Datasets were collected from the GEO (Barrett et al, 2013), the ENCODE Consortium (Davis et al, 2018), and the National Institute of Health Roadmap Epigenomics Consortium (Bernstein et al, 2010). Raw sequenced reads were downloaded from the NCBI SRA (Leinonen et al, 2011).

### Integration of user-derived sequencing data

Users can integrate local data to explore private data without having to install additional software, and visually compare unpublished with published data. *Bigwig* files can be loaded to display signal coverage, *bed* files for genomic features such as peaks or chromatin states and pairwise interactions, and *hic* files (Durand et al, 2016) for genome contact maps. The visualization of user-submitted data is restricted to the client side: files are loaded from the local file system to NAVi but not uploaded to QC Genomics servers. Note that in the present release, user data cannot be used in queries involving qcChromStater or qcComparator.

### Processing of sequencing data

Raw sequence files in the SRA format were converted to FASTQ (Cock et al, 2010) using the NCBI SRA Toolkit. For Hi-C, ChIP-seq, and related experiments, reads were aligned to the reference genome assembly using Bowtie2 (Langmead & Salzberg, 2012) with the "very-sensitive" local preset. RNA-seq reads were aligned with STAR (Dobin et al, 2013) using the "outSAMmultNmax 1", "outFilterMismatchNmax 10", and "twopassMode Basic" flags. The genome assemblies used

were hg19 (*Homo sapiens*), mm9 (*Mus musculus*), dm3 (*Drosophila melanogaster*), sacCer3 (*Saccharomyces cerevisiae*), ce10 (*Caenorhabditis elegans*), TAIR10 (*Arabidopsis thaliana*), danRer7 (*Danio rerio*), galGal4 (*Gallus gallus*), rn5 (*Rattus norvegicus*), and panTro4 (*Pan troglodytes*). Aligned reads were stored in the BAM format using SAMtools (Li et al, 2009).

For RNA-seq, ChIP-seq, and other enrichment-related experiments, global and local quality indicators were inferred by NGS-QC Generator and read coverage was calculated using in-house scripts to generate bigWig files (Pigeon pipeline). For proximity-ligation experiments such as Hi-C, genome contact maps were generated using LOGIQA (Mendoza-Parra et al, 2016). In all cases, collected datasets were paired to their corresponding metadata available in GEO and indexed in a database to generate an efficient query system. All aforementioned in-house scripts are publicly available here: https://github.com/SysFate/Pigeon.

### Browsing with NAVi

NAVi displays search results in a table of datasets which match the queried terms. Metadata annotations and analysis results are shown for each experiment to distinguish different experimental conditions and quality assessments. Detailed metadata annotations are accessible by clicking the button on the right of the table entry, and links to the repository of origin and publications are provided. By default, experiments are sorted according to the date they were submitted to the repository (most recent first), but users can sort results by any column. In addition, users can filter results by using individual column search boxes, allowing for complex search options. Users can browse results and select experiments of interest. Selected experiments can be sent to qcComparator for assessing a global enrichment patterns comparison or to qcChromStater to identify co-occurring patterns and infer functional annotations. Selected experiments can also be visualized in NAVi's web-based genome browser, which provides a gene/genome coordinate–centric query panel.

### Visualizing genomic features with NAVi

NAVi visualizes enrichment-based genomic data, such as ChIP-seq, and chromosome conformation data, such as Hi-C, in a web-based genome browser. Enrichment-based experiments were processed with NGS-QC Generator to monitor local quality (local QC, covering 500 bp regions). Local quality for a region corresponds to the extent of changes in its enrichment amplitude when only 50% of randomly selected mapped reads of the entire dataset are used for reconstructing the profile (also referred to as robustness); thus, regions of highest quality display the smallest changes (Mendoza-Parra et al, 2013b). The NGS-QC Generator database hosts local Qc tracks to display high-quality regions as heat maps for all qualified ChIP-seq and related entries.

Genome coverage tracks visualize signal data by calculating the read coverage, that is, the number of mapped reads per bin, where bins are consecutive, fixed-size windows. They are available for all enrichment-based experiments, such as ChIP-seq, RNA-seq, or chromatin accessibility and are displayed as area graphs. A common step in analysis workflows is the removal of duplicated reads

 **Life Science Alliance**

(presenting the same start and end position, also called "clonal reads"). In NAVi, the Genome coverage tracks can display signal data with or without duplicate removal; this allows users to monitor the effect of such putative PCR-based artifacts.

Binding events, and accessible chromatin regions were identified with MACS2 (Zhang et al, 2008) for ChIP-seq and chromatin accessibility experiments. Called peaks can be visualized as colored blocks, where the degree of darkness corresponds to the peak *P*-value. However, users should critically assess this information, as peak calling annotation has been performed only with the MACS2 peak caller and default parameters. In fact, different types of binding profiles, such as broad-peak H3K27me3 or sharp-peak TF profiles require specific tuning of parameters and possibly the use of dedicated peak callers (Mendoza-Parra et al, 2013a).

Because of their enormous complexity, the integration and visualization of chromatin architecture experiments, such as Hi-C, remains a challenge. Genome interaction maps, imported from LOGIQA, were analyzed at four resolutions (5, 25, 100 kb, and 1 Mb) allowing for exploration of long-range interactions at multiple scales. Contact maps can be represented as heat maps or arcs: the resolution is automatically selected based on the genomic range when displaying chromatin interactions as heat maps (5 or 25 kb for arc visualization), thus allowing for a detailed view of genomic interactions. Users can change the color score of heat maps and filter interactions by contact frequency and locus.

Users can save their current NAVi session (displayed experiments and customized views) and share the session link with colleagues.

### qcComparator

qcComparator performs global enrichment comparisons over up to 500 datasets. For this, the tracks of local quality-controlled region (local QC) for all qualified datasets were binarized (presence or absence of local enrichment per 500-nt region bin) and pairwise compared by computing similarity indexes. Two distance metrics were systematically computed for each pair:

$$Tanimoto\ Index = 1 - \frac{(A \cap B)}{(A \cup B) - (A \cap B)},$$

$$Dice\ index = 1 - \frac{2 * (A \cap B)}{(A \cup B)},$$

where A and B correspond to the binarized local QC tracks.

To minimize the time from query to display of the results, all pairwise similarity distance metrics have been precomputed. qcComparator displays similarity indexes in a heat map square matrix format in which multiple intuitive functionalities were incorporated, including a zoom-in option and the possibilities to highlight target molecules, cell/tissue types, or quality score–associated clusters. Currently, up to 500 datasets can be displayed in the similarity matrix and selected clusters can be redirected to either NAVi or qcChromStater.

For larger numbers of datasets, as illustrated in Fig 3, we have used the t-SNE strategy (van der Maaten & Hinton, 2008); this option is currently not available but will be incorporated in a future release.

### qcChromStater

qcChromStater uses the binarized local quality-controlled region (local QC) tracks of queried datasets (500-nt region bins). In contrast to qcComparator, it performs a comparison of queried datasets upon request, which is highly computation demanding. Therefore, this analysis is currently limited to 100 datasets.

The qcChromStater algorithm crosses all binarized tracks and assigns a combinatorial state identifier per genomic region. Finally, the frequency of combinatorial state IDs over the entire genome is computed to evaluate their global enrichment. By default, only the 100 most frequent states are displayed, but users can further filter results by defining a minimum fraction of local QC regions per state. qcChromStater merges, by default, local QC tracks associated with multiple datasets related to the same target molecule. This option is unable when studying different cell/tissue types.

The relevant combinatorial states are shown together with the enrichment coverage per associated target molecule. In addition, a heat map displays the stratification of each state in the context of the genomic landscape (TSS, gene body, TES, and proximal/distal regulatory element, by default corresponding to a distance of 10 and 50 kb of the TSS, respectively). Each state can be described by the user with a functional annotation. In all cases (combinatorial states and their related stratifications), users have the possibility to access the list of annotated genes.

## Supplementary Information

## Acknowledgements

We thank Guillaume Seith and Julien Seiler from the Institut de Génétique et de Biologie Moléculaire et Cellulaire informatics platform for their technical support, in particular the setup and maintenance of the required computational devices. These studies were supported by funds to our team from the Plan Cancer 2009–2013, AVIESAN-ITMO Cancer, the Ligue National Contre le Cancer (Equipe Labellisée), the Fondation Recherche Medicale, Conectus Alsace, and the Institut National du Cancer (INCa). Currently, the qcGenomics project is supported by the "Genopole Thematic Incentive Actions" (ATIGE-2017) and the institutional bodies Commissariat à l'Energie Atomique, Centre National de la Recherche Scientifique, and University Evry-Val d'Essonne.

### Author Contributions

M Blum: data curation, software, investigation, and writing—original draft, review, and editing.
P-E Cholley: data curation, software, investigation, and writing—original draft, review, and editing.
V Malysheva: validation, visualization, and methodology.
S Nicaise: data curation, software, validation, and investigation.
J Moehlin: software and validation.

H Gronemeyer: funding acquisition, project administration, and writing—original draft, review, and editing.

MA Mendoza-Parra: conceptualization, supervision, funding acquisition, investigation, project administration, and writing—original draft, review, and editing.

## Conflict of Interest Statement

The authors declare that they have no conflict of interest.

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
