## [Reviewer comments · Life Science Alliance]

Life Science Alliance

A comprehensive resource for retrieving, visualizing and integrating functional genomics data

Matthias Blum, Pierre-Etienne Cholley, Valeriya Malysheva, Samuel Nicaise, Julien Moehlin, Hinrich Gronemeyer, and Marco Mendoza-Parra

DOI: <https://doi.org/10.26508/lsa.201900546>

Corresponding author(s): Marco Mendoza-Parra, Institut Francois Jacob; University Evry-val-d'Essonne

Review Timeline:	Submission Date:	2019-09-10
	Editorial Decision:	2019-10-02
	Revision Received:	2019-11-12
	Editorial Decision:	2019-11-26
	Revision Received:	2019-11-27
	Accepted:	2019-11-28

Scientific Editor: Andrea Leibfried

Transaction Report:

October 2, 2019

Re: Life Science Alliance manuscript #LSA-2019-00546-T

Marco Antonio MENDOZA-PARRA
Institut Francois Jacob; University Evry-val-d'Essonne

Dear Dr. MENDOZA-PARRA,

Thank you for submitting your manuscript entitled "A comprehensive resource for retrieving, visualizing and integrating functional genomics data" to Life Science Alliance. The manuscript was assessed by expert reviewers, whose comments are appended to this letter.

As you will see, the reviewers tested and appreciate qcGenomics. They provide constructive input on how to render the suite more user-friendly and on how to improve its value to users. We would thus like to invite you to submit a revised version of your manuscript and a re-worked suite, incorporating the reviewer input.

Thank you for this interesting contribution to Life Science Alliance. We are looking forward to receiving your revised manuscript.

Sincerely,

Andrea Leibfried, PhD
Executive Editor
Life Science Alliance
Meyerhofstr. 1

69117 Heidelberg, Germany
t +49 6221 8891 502
e a.leibfried@life-science-alliance.org
www.life-science-alliance.org

B. MANUSCRIPT ORGANIZATION AND FORMATTING:

Reviewer #1 (Comments to the Authors (Required)):

Blum et al. described a tool suite providing a comprehensive resource for retrieving, visualizing and integrating functional genomics data. The tool is fully accessible and the friendly interface is very easy to use. The tool allow for rapid visualization of many datasets at the same time providing an easy way to gain insights into the quality of the genomics resources, visualization of the epigenomic landscape of selected genes or testing hypothesis about transcription factor binding. As such, the tool will be very useful for all the scientific community interested in gene regulation.

Specific remarks

1. Currently only the sample ID appears on the genome browser. It will be very convenient to be able to rename the samples.
2. Pre-processed data should also include datasets from the Blueprint consortium.
3. In figure 4C it is indicated "gene coverage" and "transcript coverage" to indicate the same thing. However, what the panel actually shows is the coverage for the regions defined in 4B, not necessarily the transcript itself. I suggest to replace gene/transcript by "region" coverage.
4. When citing the tools for visualization of functional genomics data, the ReMAP tool should also be cited (Griffon, et al. Nucleic Acids Res 2014.)

Typos

P6: a several descriptors -> several descriptors

P15: modifications patterns -> modification patterns

Reviewer #2 (Comments to the Authors (Required)):

Blum et al propose here a new online application suite, qcGenomics, that allows to mine, visualize and analyse publicly available genomics data sets. The new application integrates within an existing platform (ngs-qc.org) featuring a quality-assessment database of more than 82000 publicly available data sets. The search interface (NAVi) is user-friendly and allows for easy identification of relevant data sets. From here, the user can select datasets to visualize them in an embedded genome browser, compare them in a heatmap or execute a chromatin state finder. The paper is generally well writing and easy to follow. The exposed features are of common interest and definitely worth publication. Nevertheless, I feel like a number of fixes/improvements are necessary before publication.

*** Major remarks

- NGS-qc is definitely a valuable resource. Unfortunately the data is a bit outdated and should be re-analyzed against recent genome build e.g. dm6, hg38, mm10, rn6... Without this, it is not possible to visualize user own unpublished data together with public data which is one of the most interesting aspect the platform.
- The behavior of the genome browser is unstable. Looking for „atac" and post-filtering for „dm3", I blindly selected 6 „embryo" datasets in the first page. Moving to the genome browser invariably resulted in "Missing tracks. 6 tracks could not be loaded.". I try with other datasets and this error actually occurs frequently, what is going on here? This should be fixed or the datasets should not be listed.
- By default, the genome browser loads the δ RChI heatmap tracks. For enrichment-based datasets, I think most of the users want to visualize the coverage tracks. It is unfortunately quite tedious to go over each track label and right-click to activate the coverage track. Controls are needed to easily perform these operations in batch. Moreover, the visualization of input data is puzzling to me. Why are there so many nice regions visible in the δ RChI heatmap tracks while no peaks are present (eg GSM2199208 input chip-seq ; dm3:chr2R:19,533,892-19,785,700) ?

- Upload of bigwig files errored with message "Wrong chromosome size for chr21 in dm3 assembly." and the upload of 4 column bed files errored with message "Line 1: invalid format.". These files perfectly load in other genome browsers like IGV. Bigwig support should be fixed and I'd recommend to extend BED support to the 4 column format as well.
- qcChromStater and QC Comparator are not available for dm3 (at least).
- The methods section must be globally reviewed to bring the language quality up to publication standards. For example, p19 first sentence ("Then NGS-QC ...") or the qcChromStater section.

*** Important remarks

- The database does not integrate datasets published in EBI's array express. Authors should really consider EBI's array express as a second source.
- After (un)selecting datasets from the NAVi interface, the genome browser is not updated and one needs to perform an action. Could the refresh occur automatically?
- NAVi allows to filter datasets, select a few then use another filtering criteria and select more datasets. This is great but there is no easy way to list all selected datasets to review them or remove some. One can remove datasets in the genome browser but it is not the easiest place to review datasets (labels are truncated and no dataset information is visible)
- It seems that NAVi allows to search by e.g. pubmed ID. Unfortunately the table does not display much information like publication references. It is therefore unclear how to quickly check what the search returned (clicking the 'info' icon is not an option to review dozens of datasets). It would be good to enrich this dataset table to integrate all available information.
- QC Comparator: it would help to add labels for datasets (X and Y)
- I suggest to support the cooler format for HiC
- In the methods, the authors mention that computation include the use of in house scripts. The data processing workflows should be more transparent and explained in details. I suggest to publish the workflows and custom code in e.g. github.

*** Minor:

- I encourage the authors to support the integration of user provided data in qcChromStater and qcComparator. I believe this is one of the most relevant use cases for researchers.
- QC Comparator: why is the "High" on the left of the slider? This is counter intuitive.
- Documentation videos don't play on safari
- p7 line 2: "...is apparent in at..." => : "...is apparent at..."
- p14 line 12 : "update" => "updated"

Re: Life Science Alliance manuscript #LSA-2019-00546-T « A comprehensive resource for retrieving, visualizing and integrating functional genomics data”

letter addressing the reviewers' comments point by point.

Reviewer #1 (Comments to the Authors (Required)):

Blum et al. described a tool suite providing a comprehensive resource for retrieving, visualizing and integrating functional genomics data. The tool is fully accessible and the friendly interface is very easy to use. The tool allow for rapid visualization of many datasets at the same time providing an easy way to gain insights into the quality of the genomics resources, visualization of the epigenomic landscape of selected genes or testing hypothesis about transcription factor binding. As such, the tool will be very useful for all the scientific community interested in gene regulation.

We are pleased to hear that reviewer #1 emphasizes the usefulness of our tool for the scientific community, and that the qcGenomics suite provides not only an easy way to interrogate the quality and visualize large amounts of public datasets but also simplifies the testing of TF binding hypothesis.

We are aware of several possibilities to further improve qcGenomics, some of which are mentioned by the reviewer and are addressed below, but keeping in mind that maintaining such database over the years will require further releases, we aim at this state to focus on the major aspects and delay small modifications for the future.

Specific remarks

1. Currently only the sample ID appears on the genome browser. It will be very convenient to be able to rename the samples.

We do not quite understand the reviewer's suggestion. Please note that the label of each track contains information concerning the cell line/tissue, the type of target molecule (e.g. H3K27me3), and the experiment (e.g. histone mark). While technically feasible, we do not consider optimal to allow users modify the displayed sample ID, mainly to avoid issues of accuracy when users export such displays. Hence, users are responsible of modifying the label in a second time when editing their taken screen shots.

2. Pre-processed data should also include datasets from the Blueprint consortium.

We fully agree with this suggestion. However, qcGenomics requires to have access to the raw data (i.e. fastq files) or aligned files (BAM/SAM or BED format) for performing quality assessment. In the early phases of the Blueprint project only processed files could be accessed from Blueprint and our request for raw files was denied by the Blueprint Consortium. However, in the meantime several Blueprint datasets have been deposited in GEO and are part of the qcGenomics database (e.g. PMID: 26673693).

3. In figure 4C it is indicated "gene coverage" and "transcript coverage" to indicate the same thing. However, what the panel actually shows is the coverage for the regions defined in 4B, not necessarily the transcript itself. I suggest to replace gene/transcript by "region" coverage.

We have corrected these captions in the revised manuscript. We now refer in all cases to "**transcript coverage**", because as indicated on the related documentation (<http://ngs-qc.org/chromstater/index.php#documentation>), the coverage (in %) calculated by qcChromStater is based on the number of transcripts (not the number of reported gene names).

4. When citing the tools for visualization of functional genomics data, the ReMAP tool should also be cited (Griffon, et al. Nucleic Acids Res 2014.)

Please note that we have cited the latest updated version of REMAP (Cheneby et al. 2018) in the discussion. We have added the initial version of 2014 in the revised version of the manuscript.

Typos

P6: a several descriptors -> several descriptors

P15: modifications patterns -> modification patterns

We Thank the reviewer for pointing out these typos; all have been corrected in the revised version of the manuscript.

Reviewer #2 (Comments to the Authors (Required)):

Blum et al propose here a new online application suite, qcGenomics, that allows to mine, visualize and analyse publicly available genomics data sets. The new application integrates within an existing platform (ngs-qc.org) featuring a quality-assessment database of more than 82000 publicly available data sets. The search interface (NAVi) is user-friendly and allows for easy identification of relevant data sets. From here, the user can select datasets to visualize them in an embedded genome browser, compare them in a heatmap or execute a chromatin state finder.

The paper is generally well writing and easy to follow. The exposed features are of common interest and definitely worth publication. Nevertheless, I feel like a number of fixes/improvements are necessary before publication.

We thank reviewer #2 for her/his appreciation of the qcGenomics suite and the manuscript.

*** Major remarks

- NGS-qc is definitely a valuable resource. Unfortunately the data is a bit outdated and

should be re-analyzed against recent genome build e.g. dm6, hg38, mm10, rn6... Without this, it is not possible to visualize user own unpublished data together with public data which is one of the most interesting aspect the platform.

While we understand this concern, reviewer #2 may underestimate the enormous effort to re-align > 80,000 datasets to a new reference genome. Indeed, if the user is interested in comparing his/her own data with the content of our database, realigning few files on their side is straight forward and requires dramatically less effort than reprocessing the entire database each time a new reference genome becomes available.

- The behavior of the genome browser is unstable. Looking for „atac" and post-filtering for „dm3", I blindly selected 6 „embryo" datasets in the first page. Moving to the genome browser invariably resulted in "Missing tracks. 6 tracks could not be loaded.". I try with other datasets and this error actually occurs frequently, what is going on here? This should be fixed or the datasets should not be listed.

We apologize for this error message. The reason for this message is that qcGenomics can provide global quality scores for several organisms but the required files for genome browser displays were preferentially generated for human and mouse-related datasets. This being said, by following the example suggested by reviewer #2, querying for ATAC-seq assays and further selecting for “dm3”, qcGenomics provides global quality scores for 48 datasets, from which half of them can be visualized with the genome browser. We plan in a further release, to increase the number of “fully processed files” for all organisms such that users could visualize enrichment patterns for most (if not all) qualified data retrieved in our database.

- By default, the genome browser loads the δ RICI heatmap tracks. For enrichment-based datasets, I think most of the users want to visualize the coverage tracks. It is unfortunately quite tedious to go over each track label and right-click to activate the coverage track. Controls are needed to easily perform these operations in batch.

There are two main reasons for displaying dRICI heatmap tracks by default instead of coverage tracks. (i) qcGenomics is a database focused on displaying qualified datasets and (ii) it is easier to visualize a large number of tracks as “heatmap” rather than coverage track. This being said, we have included as part of this revised version an option for displaying all coverage tracks in batch.

Moreover, the visualization of input data is puzzling to me. Why are there so many nice regions visible in the δ RICI heatmap tracks while no peaks are present (eg GSM2199208 input chip-seq ; dm3:chr2R:19,533,892-19,785,700) ?

The sample cited by reviewer #2 corresponds to an input sample for a drosophila assay. Indeed, for small genomes, our strategy for assessing local qualified regions (dRICI) is biased by the high coverage (> 29 million reads for GSM2199208), mainly due to the used background model (Poisson distribution). This issue is not observed for larger genomes like mouse or human, with the exception of input control samples presenting an “enrichment-like pattern” which is indicative of improper sample preparation. For future releases in which smaller genomes will be available for visualization, we will enhance the background model by incorporating more appropriated distributions (e.g. negative binomial distribution).

- Upload of bigwig files errored with message "Wrong chromosome size for chr2I in dm3 assembly." and the upload of 4 column bed files errored with message "Line 1: invalid

format.". These files perfectly load in other genome browsers like IGV. Bigwig support should be fixed and I'd recommend to extend BED support to the 4 column format as well.

We are grateful to the reviewer for pointing out these errors. We have fixed the errors with the Bigwig format and specified the supported file formats associated to their corresponding usage.

- qcChromStater and QC Comparator are not available for dm3 (at least).

Indeed, qcGenomics downstream analysis tools are only available for human and mouse-related datasets. As indicated before, we will add this option in a future update.

- The methods section must be globally reviewed to bring the language quality up to publication standards. For example, p19 first sentence ("Then NGS-QC ...") or the qcChromStater section.

Following this remark, the methods section has been reviewed in the revised manuscript.

*** Important remarks

- The database does not integrate datasets published in EBI's array express. Authors should really consider EBI's array express as a second source.

As indicated in the comments for reviewer #1; qcGenomics requires to have access to the raw data (i.e. fastq files) or aligned files (BAM/SAM or BED format) for performing first quality assessment. Please note that ArrayExpress and GEO have been working together to synchronize their content, such that today most of the ArrayExpress entries are retrieved in GEO. However, we will consider processing data available in ArrayExpress not available in GEO. Currently, this concerns only 1,687 mouse and human datasets.

- After (un)selecting datasets from the NAVi interface, the genome browser is not updated and one needs to perform an action. Could the refresh occur automatically?

We have incorporated this suggested modification as part of our current release.

- NAVi allows to filter datasets, select a few then use another filtering criteria and select more datasets. This is great but there is no easy way to list all selected datasets to review them or remove some. One can remove datasets in the genome browser but it is not the easiest place to review datasets (labels are truncated and no dataset information is visible)

At this stage, only a manual selection for removing datasets is available. We may consider this suggestion for a further release.

- It seems that NAVi allows to search by e.g. pubmed ID. Unfortunately the table does not display much information like publication references. It is therefore unclear how to quickly check what the search returned (clicking the 'info' icon is not an option to review dozens of

datasets). It would be good to enrich this dataset table to integrate all available information.

We may consider this suggestion for a further release with modified display options. In the current display adding information concerning the related articles will “over-crowd” the presentation. Given that users have all links to retrieve the related reference we do not consider this as a major problem.

- QC Comparator: it would help to add labels for datasets (X and Y)

In fact, qcComparator is able to display up to 500 different datasets at once; thus, it is technically difficult to display such number of labels beside the displayed matrix. This being said, when users pass with the mouse pointer over the dynamic matrix, the identity for each of the datasets (in X and Y) is displayed. Furthermore, the table at the bottom of the matrix is displaying the order of the dataset in the X and Y axis.

- I suggest to support the cooler format for HiC

In fact, as far as we know, the cooler format cannot be parsed on the user side. Hence, its usage as part of qcGenomics might require important resources in the back-end; reason why we do not consider to support this format.

- In the methods, the authors mention that computation include the use of in-house scripts. The data processing workflows should be more transparent and explained in details. I suggest to publish the workflows and custom code in e.g. github.

Following the reviewer’s suggestion, we have deposited the in-house scripts for downloading/processing public data (termed ‘Pigeon’) at GitHub. These scripts are available here: <https://github.com/SysFate/Pigeon>

*** Minor:

- I encourage the authors to support the integration of user provided data in qcChromStater and qcComparator. I believe this is one of the most relevant use cases for researchers.

While we do understand this request, for processing users’ data in qcChromStater or qcComparator, we have to access to their raw files or aligned files, from which quality descriptors might be inferred for comparisons. This step is rather time-consuming, thus at this stage we do not have a simple solution for supporting this option.

- QC Comparator: why is the "High" on the left of the slider? This is counter intuitive.

We apologize for this counter intuitive annotation. In fact, the proper terminology is “dissimilarity index”, with low dissimilarity on the left and high in the right side.

- Documentation videos don't play on safari

We have addressed this issue by including a link to access to the videos via “Youtube”.

- p7 line 2: "...is apparent in at..." => : "...is apparent at..."
- p14 line 12 : "update" => "updated"

Thank you for pointing out these typos which have been corrected in the revise versionof the manuscript.

November 26, 2019

RE: Life Science Alliance Manuscript #LSA-2019-00546-TR

Dr. Marco Antonio MENDOZA-PARRA
Institut Francois Jacob; University Evry-val-d'Essonne
2, Gaston Crémieux
Evry 91000
France

Dear Dr. MENDOZA-PARRA,

Thank you for submitting your revised manuscript entitled "A comprehensive resource for retrieving, visualizing and integrating functional genomics data". As you will see, reviewer #2 appreciates the introduced changes, and we would thus be happy to publish your paper in Life Science Alliance pending final revisions necessary to meet our formatting guidelines:

- please provide the link to your tool in the abstract
- the suppl figure files will be displayed in-line in the HTML version of the manuscript; please upload them as individual files, the suppl figure legends can get moved into the main manuscript file

A. FINAL FILES:

-- Summary blurb (enter in submission system): A short text summarizing in a single sentence the study (max. 200 characters including spaces). This text is used in conjunction with the titles of papers, hence should be informative and complementary to the title. It should describe the context and significance of the findings for a general readership; it should be written in the present tense

and refer to the work in the third person. Author names should not be mentioned.

B. MANUSCRIPT ORGANIZATION AND FORMATTING:

Sincerely,

Reviewer #2 (Comments to the Authors (Required)):

The authors have addressed most of my issues. The resource remains quite limited for *Drosophila* researchers, but I believe this is useful for the mouse/human community.
So I therefore recommend to publish the paper.

November 28, 2019

RE: Life Science Alliance Manuscript #LSA-2019-00546-TRR

Dr. Marco Antonio MENDOZA-PARRA
Institut Francois Jacob; University Evry-val-d'Essonne
2, Gaston Crémieux
Evry 91000
France

Dear Dr. MENDOZA-PARRA,

Thank you for submitting your Resource entitled "A comprehensive resource for retrieving, visualizing and integrating functional genomics data". It is a pleasure to let you know that your manuscript is now accepted for publication in Life Science Alliance. Congratulations on this interesting work.

DISTRIBUTION OF MATERIALS:

Again, congratulations on a very nice paper. I hope you found the review process to be constructive and are pleased with how the manuscript was handled editorially. We look forward to future exciting submissions from your lab.

Sincerely,
